# Probiotics, Prebiotics and Synbiotics—A Promising Strategy in Prevention and Treatment of Cardiovascular Diseases?

**DOI:** 10.3390/ijms21249737

**Published:** 2020-12-20

**Authors:** Beata Olas

**Affiliations:** Department of General Biochemistry, Faculty of Biology and Environmental Protection, University of Lodz, Pomorska 141/3, 90-236 Lodz, Poland; beata.olas@biol.uni.lodz.pl; Tel./Fax: +48-42-635-4484

**Keywords:** probiotic, prebiotic, synbiotic, cardiovascular disease

## Abstract

Recent evidence suggests that probiotics, prebiotics and synbiotics may serve as important dietary components in the prevention (especially) and treatment of cardiovascular diseases (CVD), but the recommendations for their use are often based on brief reports and small clinical studies. This review evaluates the current literature on the correlation between CVD and probiotics, prebiotics and synbiotics. Although research on probiotics, prebiotics and synbiotics has grown exponentially in recent years, particularly regarding the effect of probiotics on CVD, their mechanisms have not been clearly defined. It has been proposed that probiotics lower cholesterol levels, and may protect against CVD, by increasing bile salt synthesis and bile acid deconjugation. Similar effects have also been observed for prebiotics and synbiotics; however, probiotics also appear to have anti-oxidative, anti-platelet and anti-inflammatory properties. Importantly, probiotics not only have demonstrated effects in vitro and in animal models, but also in humans, where supplementation with probiotics decreases the risk factors of CVD. In addition, the properties of commercial probiotics, prebiotics and synbiotics remain undetermined, and further experimental research is needed before these substances can be used in the prevention and treatment of CVD. In particular, well-designed clinical trials are required to determine the influence of probiotics on trimethylamine-N-oxide (TMAO), which is believed to be a marker of CVDs, and to clarify the long-term effects, and action, of probiotic, prebiotic and synbiotic supplementation in combination with drug therapy (for example, aspirin). However, while it cannot be unequivocally stated whether such supplementation yields benefits in the prevention and treatment of CVDs, it is important to note that clinical studies performed to date have not identified any side-effects to use.

## 1. Introduction

Cardiovascular diseases (CVD) are conditions that involve the heart or blood vessels, the most well-known being coronary artery diseases (angina and myocardial infarction), stroke, hypertensive heart diseases, cardiomyopathy, venous thrombosis, arrhythmia and thromboembolic disease. The underlying mechanisms vary depending on the disease. For example, coronary artery disease, peripheral artery disease, and stroke involve atherosclerosis [1,2]. They may be associated with various lifestyle factors such as poor diet or smoking, and are characterized by high blood pressure, high cholesterol levels, obesity and hyperactivation of blood platelets [1]. CVDs are believed to have been responsible for about 18 million deaths worldwide (about 32%) in 2015 [2].

CVD management does not necessarily require therapeutic intervention or prophylaxis. For example, it is known that a diet high in fruits and vegetables can decrease the risk of CVD via various mechanisms, including reducing plasma cholesterol level and inhibiting blood platelet activation. However, such dietary modification can also result in changes in the microbial and metabolic composition of the intestines, which themselves are known to be characteristic of a range of physiological and pathological states, including CVD; the composition of the gut microbiota has been found to be associated with atherogenesis, thrombosis, chronic heart conditions and arterial hypertension [3,4,5,6,7], and this correlation has also been noted in a recent meta-analysis [8]. 

Dietary supplements, often termed functional foods, can bestow beneficial effects against various risk factors associated with cardiovascular diseases. However, little, if anything, is known about the role of probiotic, prebiotic and symbiotic supplements as important dietary components in the prevention and treatment of CVD. For example, in a review paper, Vascquez et al. [9] indicated that probiotics may decrease the production of reactive oxygen species, and reduce oxidative stress. Nevertheless, some brief reports and small clinical studies have been conducted, and these may nevertheless offer new directions in treatment; in addition, the body of research regarding these supplements, particularly concerning the effect of probiotics on cardiovascular diseases, has recently grown exponentially. For example, 201 articles were published during 2014–2020, and 63 were published on “probiotics and cardiovascular diseases” during 2009–2013 (PubMed June, 2020). Therefore, the aim of this review is to discuss the roles of probiotics, prebiotics and synbiotics in the prevention and treatment of cardiovascular diseases based on data from journals recorded in international databases, including *inter alia* PubMed and Scopus.

## 2. Probiotics

Belief in the beneficial effects of certain types of bacteria on the human body dates back to ancient times. It is known that Plinius the Elder advocated drinking beverages made from fermented milk to alleviate stomach troubles [10]. 

The term *probiotic* is derived from the Greek *pro bios*, meaning *for life*. It was introduced in 1965 by Lilly and Stillwell, who identified a probiotic as “an organism or substance that affects the balance of intestinal microflora”. In 1989, Fuller saw the benefits of probiotics and described them as a “live, microbiological food supplement” [10]. In 2002, the Food and Agriculture Organization/World Health Organization (WHO) defined probiotics as “live micro-organisms which when present in adequate amounts, confer benefits to the body of the host” [11].

In the early twentieth century, research on probiotics chiefly focused on the species of *Lactobacillus* found in dairy products, for example *Lactobacillus casei* (the first probiotic strain isolated in Japan), and on its Western equivalent *Lactobacillus acidophilus*. In 1970, Ilyich Metchnikoff (who is usually considered the founder of probiotics) reported that the quantity of *Lactobacilli* in the gastrointestinal tract has an influence on the health of the host [12]. *Lactobacillus* is one of the main genera of probiotic microorganisms with a long history of safe use; it is recognized by the US Food and Drug Administration (FDA) for consumption, and is approved for use in dairy products [10,13].

Probiotic bacteria are constantly present in humans and are naturally present in the digestive tract and the mouth; they are consumed around the world along with traditional food products. They have also been widely used to preserve the freshness of foods such as salami, cheese, soy sauce or sauerkraut for hundreds, or even thousands, of years [10].

Probiotics are believed to adhere to intestinal epithelial cells, preventing the development of harmful microorganisms by producing anti-bacterial substances including bacteriocins and organic acids. The quantitative share of bacteria in the digestive tract is variable, with the number of *Bifidobacteria* in the human colon ranging from 8 × 10^4^ to 2.5 × 10^13^ cells (mean 1.6 × 10^10^ cells), and the number *Lactobacillus* bacteria ranging from 4 × 10^4^ to 3.2 × 10^12^ (mean 4 × 10^9^ cells). Twelve species of *Bifidobacteria* are known to occur in humans, particularly *B. longum*, *B. pseudolongum*, *B. animalis* and *B. bifidum*; *Bifidobacteria* is one of the first genera of microorganisms to colonize the sterile digestive tracts of newborns, and is believed to constitute the largest group of intestinal microflora, representing up to 95% of the total microbial population during breast feeding [10]. 

WHO regulations state that the number of living cells in probiotic foods at the time of human consumption may not be lower than 10^6^ cells per 1 mL or 1 g of product, while the therapeutic dose is 10^8^–10^9^ cells in 1 mL or 1 g of product. Given that the microorganism must remain viable and should not be killed by passing through the gastrointestinal tract, the probiotic must be resistant to the action of gastric juice and bile salts. After passing through this chemical barrier, probiotics should adhere to the surface of the intestine, where their health-promoting functions can be realized. In addition, if probiotic microorganisms do not colonize the intestine, the repeated consumption of probiotic products may be necessary [10].

Probiotics can enhance the nonspecific cellular immune response through the activation of natural killer cells, macrophages and the release of various cytokines. They can also improve the gut mucosal immune system by increasing the number of IgA(+) cells [14,15]. In addition, probiotics can also aid the process of digestion and the breakdown of lactose, improve the absorption of minerals such as calcium, zinc, iron and manganese, and improve the synthesis of many vitamins, including thiamin, riboflavin, niacin, pantothenic acid and vitamin K. Probiotics play an important role in the treatment of various diseases, such as hepatic disease, diarrhea and gastroenteritis. Probiotics have also been shown to have anti-proliferative, pro-apoptotic and anti-oxidative properties [14,15]. 

### Probiotics and CVDs

Probiotics may support the prevention and treatment of some cardiovascular diseases by reducing elevated cholesterol levels [10,16,17]. Unfortunately, most studies on the effects of probiotics on cardiovascular diseases have been performed in vitro. While some in vivo studies have examined the effect of probiotic supplementation on risk factors of CVDs in people and animals, the number of commercial probiotic supplements is too limited to unequivocally demonstrate that such probiotics have a beneficial effect on the cardiovascular system.

Probiotics may reduce cholesterol levels by several mechanisms [18]. Most *Bifidobacteria* bacteria demonstrate higher choliloglicin hydrolase activity than other microorganisms. This enzyme hydrolyzes the amide bonds conjugated with taurine or glycine in bile acids, resulting in the release of primary bile acids; these are easily precipitated at low pH, resulting in their expulsion from the gastrointestinal tract. As these are not reabsorbed from the intestine, they must be replaced by bile produced in the liver from blood cholesterol. Alternatively, *Bifidobacteria*-driven cholesterol assimilation, or precipitation in the acidic environment of the gut, can also reduce plasma cholesterol level by slowing its absorption from food into the blood [10]. 

A recent meta-analysis by Mo et al. [19] indicated that the use of probiotics significantly lowers total cholesterol and low-density lipoprotein (LDL) cholesterol in hypercholesterolemic adults. Niamah et al. [20] also note that soy milk fermented (3 mL/day) by a probiotic starter (Log.11.1 cfu/mL) consisting of *S. thermophilis*, *L. acidophilus* LA-5 and *B. bifidum* BG-12 decreases the level of cholesterol and triglyceride compared with control.

Malik et al. [21] report that *L. plantarum* 299v (Lp299v) supplementation improves vascular endothelial function and reduces inflammatory biomarkers in men with stable coronary artery disease. In this experiment, twenty men with stable coronary artery disease consumed a drink containing 20 billion colony-forming units (cfu) of Lp299v per day for six weeks. Vascular endothelial function was measured by brachial artery flow-mediated dilation, and endothelial vasodilation was measured by video microscopy. It was also found that the administered probiotics can regulate the genes responsible for the intestinal transport of cholesterol (including *ABCG8*, *ABCG5* and *NPC1L1*), as well as some genes responsible for the homeostasis of cholesterol in the liver, including those controlling the production of 3-hydroxy-3-methylglutaryl-coenzyme A reductase (HMGCR) [21]. 

A recent study by Lew et al. [22] found *L. plantarum* DR7 to exert cholesterol-lowering properties via AMPK phosphorylation, resulting in the reduced expression of HMGCR. Nguyen et al. [23] also suggest that the probiotic *L. plantarum* PH40 may also possess cholesterol-lowering properties. Wang et al. [24] reported that the four-week administration of *L. acidophilus* (10^9^ cfu/mL) improves lipid-regulating activity in hypercholesterolemic rats, together with one of statin—rosuvastatin (10 mg/kg).

Systematic inflammation is commonly observed among CVDs, as are many risk factors, such as hypertension [25]. Oxidative stress is also known to play a role in the course of CVDs [24]. *Lactobacillus* and *Bifidobacterium* have been found to inhibit lipid peroxidation and ROS production, and hence may slow or even halt the development of CVDs and other oxidative stress-related diseases [26,27,28]. 

Yadav et al. [29] reported that the consumption of probiotic *L. fermentum* MTCC: 5898-fermented milk not only attenuated dyslipidemia, but also reduced oxidative stress and inflammation in male rats fed a cholesterol-enriched diet. Oxidative stress was measured by the activity of various anti-oxidative enzymes, such as catalase, superoxide dismutase and glutathione peroxidase, and the level of thiobarbituric acid reactive substances (TBARS) in the liver and kidney—a marker of lipid peroxidation. Inflammation, measured by the expression of inflammatory cytokines, including tumor necrosis factor-alpha (TNF-α) and interleukin-6 (IL-6), was also influenced at the genetic level. The study examined the effect of 90-day probiotic supplementation among male Wistar rats. The rats were divided into three groups: one was fed a standard diet (Group 1), another fed a cholesterol-enriched diet (Group 2), and another fed a cholesterol-enriched diet together with 2 mL MTCC: 5898-fermented milk (2 × 10^9^ cfu) per day (Group 3). The probiotic-fermented milk was prepared by inoculating 1% of activated *L. fermentum* MTCC: 5898 (in skimmed milk) into fresh milk (2.5% fat), which was then incubated at 37 °C for 18 h before feeding.

Tenorio-Jimenez et al. [30] report that the 12-week administration of a daily dose of 5 × 10^9^ cfu *L. reuteri* V3401 in capsules was associated with lower levels of inflammation biomarkers, such as TNF-α, IL-6, IL-8 and soluble intercellular adhesion molecule-1 (sICAM-1), in obese adults aged 18 to 65 years with metabolic syndrome, as well as with a reduced risk of CVD. Szulinska et al. [31] also observed that supplementation with the multispecies probiotic Ecologic^®^ Barrier modified both the functional and biochemical parameters of vascular dysfunction in a group of 81 obese postmenopausal Caucasian women. The subjects were assigned to three groups: (1) placebo, (2) those receiving a low dose (2.5 × 10^9^ cfu per day), and (3) those receiving a high dose (1 × 10^10^ cfu per day). The supplement was administered for 12 weeks. The authors note that the low and high doses decrease systolic blood pressure and serum vascular endothelial growth factor (VEGF), TNF-α and IL-6. Other clinical trials have also found probiotic use to be associated with a moderate or significant reduction in blood pressure among both healthy and obese participants [32,33,34,35,36]. The anti-hypertensive action of probiotics is believed to act via several mechanisms, including regulating the renin–angiotensin system [36].

The hyperactivation of blood platelets also plays a key role in thrombotic disorders. Although pharmacological strategies are available, such as those based on aspirin and its derivatives, these treatments are often accompanied by side-effects and complications that outweigh their benefits. In addition, blood platelet function is strongly influenced by diet, including the consumption of fruit and vegetables [37,38]. Although the precise roles played by probiotics in the modulation of hemostasis and its various elements, such as blood platelet function, are generally not well documented, a few in vitro and in vivo studies have nevertheless examined their effects on various elements of hemostasis, such as platelet activation. For example, Schreiber et al. reported that *L. reuteri* reduces P-selectin expression on the platelet surface and decreases blood platelet–endothelial cell interactions in rats treated with dextran sodium sulfate (DSS) [39]. In addition, Haro and Medina [40] also indicated that the oral administration of probiotic *L. casei* CRL431 may be a promising candidate for the prevention of thrombotic complications associated with pneumococcal pneumonia, especially in at-risk patients. It was found that the oral administration of CRL431 in a respiratory pneumococcal infection model in mice reduces blood platelet activation, measured by various parameters, including P-selectin expression, and modulates antithrombin levels within the normal range. These findings may indicate that *L. casei* may also influence the coagulation system [40]. In addition, in vitro experiments found no significant difference in the levels of blood platelet activation markers, such as ADP-induced platelet aggregation, between resting platelets and samples incubated with *L. rhamnosus* HN001 and *B. lactis* HN019. The authors suggest that these probiotic strains do not appear to participate in the pathogenesis of thrombotic disorders [41]. 

It is likely that inflammatory processes play an important role in the pathogenesis and progression of atherosclerosis, as elevated levels of proinflammatory cytokines have been observed in patients with atherosclerosis [42]. However, although some studies have demonstrated that probiotics can decrease the production of pro-inflammatory cytokines, their underlying mechanism remains unclear [7].

A recent study by Majewska et al. [43] found probiotic supplementation (Ecologic^®^ BARIER, containing 2.5 × 10^9^ cfu/g; for 12 weeks) to inhibit the inflammation process, reduce oxidative stress, and decrease the concentration of homocysteine in obese women (*n* = 50; aged 45–70 years).

## 3. Prebiotics

Prebiotics contain no microorganisms—only substances which stimulate their growth. These do not undergo digestion; therefore, they reach the intestinal lumen unchanged, where they can then act. For food ingredients to be considered prebiotics, they must meet certain criteria: they must be resistant to the actions of stomach acids, bile salts and other hydrolyzing enzymes in the intestine, have a known chemical structure, provide a substrate for one or more strains of beneficial bacteria, stimulate the growth and activity of the desired groups of bacteria in the digestive tract, and possess proven health benefits [10,44]. 

Prebiotics can be obtained from various sources, including breast milk, soybeans and raw oats [10,45]. However, the most popular prebiotics are the oligosaccharides contained in plants, such as asparagus, artichokes, chicory and onions [10]. Oligosaccharides may benefit the gastrointestinal tract via fermentation and the proliferation of desirable bacterial species. For example, fructo-oligosaccharide promotes the growth of bifidobacteria in vivo. Oligofructose and inulin also increase the population of colonic bifidobacteria [46,47]. Hsu et al. [48] observed that xylo-oligosaccharides and fructo-oligosaccharides affect the intestinal microbiota and precancerous development of colonic lesions in rats. Nowadays, probiotics are often added as supplements to dairy products, meat products and beverages, such as health drinks. Recently, Cherry et al. [49] reported that seaweed may be a good source of polysaccharide components, which may act as prebiotics.

### Prebiotics and CVDs

Parnell and Reiner [50] reported that probiotic supplementation lowers total serum cholesterol in a hypercholesterolemic rat model. In this experiment, rats were administered one of three diets supplemented with 0, 10 or 20% prebiotic fiber for 10 weeks. Both doses of prebiotic fiber reduced serum cholesterol concentrations by about 25%. In addition, this change was correlated with an increase in caeca digesta, as well as the up-regulation of genes involved in cholesterol biosynthesis and bile production. In addition, the obese rats with 10% prebiotic supplementation demonstrated an approximately 40% reduction in triacylglycerol accumulation in the liver (Table 1). 

Obesity is often associated with the progression of CVDs. However, both probiotic and prebiotic supplementation have been reported to have anti-obesogenic effects in various clinical trials (Table 1) [51,52,53,54,55].

The role of prebiotics in preventing and treating hypertension is reviewed in greater detail by Ghaffari and Roshanravan [56].

## 4. Synbiotics

A synbiotic is a combination of probiotics and prebiotics, typically oligosaccharides or inulin, which demonstrate an additive action and are used to restore normal bacterial flora [10]. 

### Synbiotics and CVDs

A number of studies have found synbiotics to demonstrate promising hypercholesterolemic properties [57,58,59,60]. For example, Mofid et al. [60] reported that the regular intake of synbiotic yoghurts reduces the risk of CVDs among hypercholesterolemic patients.

Haghighat et al. [59] noted that 12-week synbiotic supplementation reduces the concentration of intracellular adhesion molecule type 1 (ICAM-1), which is a risk factor for cardiovascular diseases in hemodialysis patients. The study included three groups of subjects: synbiotic, probiotic and placebo. Those in the synbiotic group received 15 g of prebiotic (three different fiber types: 5 g fructo-oligosaccharides, 5 g galacto-oligosaccharides and 5 g inulin) and 5 g probiotic powder (Bioflora^®^) containing *L. acidophilus* strain T16, *B. bifidum* strain BIA-6, *B. lactis* strain BIA-6, and *B. longum* strain LAF-5 (2.7 × 10^7^ cfu/g each) in sachets. The probiotic group received 5 g probiotics, as in the synbiotic group, with 15 g of maltodextrin powder in sachets. The placebo group received 20 g of maltodextrin powder in sachets. 

It has been found that a synbiotic containing *L. acidophilus* ATCC 4962 reduces total cholesterol, triacylglycerol, and LDL-cholesterol in hypercholesterolemic pigs, probably in the form of cholesterol esters, via the interrelated pathways of lipid transporters, including high-density lipoprotein (HDL), LDL and very low-density lipoprotein (VLDL). In this experiment, three commercially available prebiotics were used: mannitol, fructo-oligosacharides and inulin. The pigs on the symbiotic diet were supplemented with powdered *L. acidophilus* ATCC 4962 (1 g/pig per day), mannitol (1.56 g/pig per day), fructo-oligosacharides (1.25 g/pig per day) and inulin (2.2 g/pig per day). The animals were also given dietary fat (5 and 15%) (Table 1). In addition, it was found that the synbiotic reduces the deformation of erythrocytes via improved membrane permeability and fluidity [57,58].

## 5. The Effects of Probiotics, Prebiotics and Synbiotics on TMAO Levels

Trimethylamine-N-oxide ((CH_3_)_3_NO; TMAO) is an organic compound produced from choline, phosphatidylcholine, and carnitine precursors. These are converted into an intermediate compound, trimethylamine (TMA), by members of the gut microbiota, such as Acinetobacter; TMA is in turn oxidized into TMAO by the action of hepatic flavin monooxygenases (FMOs) [61]. As such, the consumption of carnitine-containing food products, such as some energy drinks and dietary supplements, results in increased TMAO levels in the blood [62]. 

However, the nature of the relationship between TMAO levels and CVDs remains unclear. Wilson et al. proposed that a high concentration of TMAO in the blood may be correlated with an increased risk of CVDs [63]. TMAO has also been indicated as a marker of cardiovascular events [64], and has been observed to influence the metabolism of cholesterol in the liver in the intestinal and artery walls. Hazen reported an increased deposition of cholesterol in peripheral cells, such as those in artery walls, in the presence of TMAO, together with the decreased removal of cholesterol [65]. In addition, carnitine intake and a high plasma concentration of TMAO have been found to correlate with low aortic lesions in ApoE -/- transgenic mice expressing cholesterol ester transfer protein (CETP) [66]. Ufnal et al. [67] and Tilg [68] suggest that TMAO may be involved in the etiology of hypertension and thrombosis, and in the regulation of arterial blood pressure. A review by Kanitsoraphan et al. [69] suggests that the concentration of TMAO could be incorporated into existing risk stratification tools, and could lead to novel prevention and treatment. 

A recent study by Hardin et al. [70] described a possible mechanism connecting TMAO and the enhancement of atherosclerotic factors. They proposed that increased TMAO concentrations in plasma promoted the upregulation of macrophage scavenger receptors, which promote the recruitment of macrophages and associated inflammation. Being unable to properly metabolize the lipids inside them, the accumulated macrophages produce foamy cells. The resulting inflammation, foamy cell production and accumulation of macrophages promote the development of atherosclerosis and other CVDs. Leustean et al. [71] described other mechanisms of TMAO action in CVDs, including blood platelet aggregation and decreases in nitric oxide level (Figure 1).

Preliminary results suggest that antibiotic therapy may suppress TMAO levels, but the long-term stability of such therapy remains unknown [64]. It has been proposed that probiotic treatment may prevent the development of CVDs following TMAO accumulation (Figure 2). However, Malik et al. [21] noted that supplementation with a drink containing *L. plantarum* 299v (Lp299v), 20 billion cfu daily for six weeks, did not significantly change plasma TMAO concentrations in a group of 20 men aged 40–75 years with stable coronary artery disease. Tripolt et al. [72] also noted that supplementation with *L. casei* Shirota (6.5 × 109 cfu, daily) three times for 12 weeks does not change TMAO levels in subjects with metabolic syndrome. Similar results were observed by Boutagy et al. [73] in a study of nineteen non-obese subjects who consumed a mix of various probiotics (900 billion live bacteria) for four weeks.

In contrast, Qui et al. [74] observed that *L. plantarum* ZDYO4 inhibits the development of TMAO-induced atherosclerosis in ApoE-/- 1.3% choline-fed mice, compared to controls. Tenore et al. [75] reported a decrease in TMAO level among a group of 90 individuals with CVD risk factors, including high total cholesterol and LDL levels, following the consumption of lactofermented annurca apple puree (125 g/day for 16 weeks). Each dose contained approximately 3 × 10^8^ cfu *L. rhamnosus* LRH11 and *L. plantarum* SGL07. 

Liu et al. reported that the consumption of probiotic-fermented purple sweet potato yoghurt may reverse congestive heart failure induced by hypertension; this was proposed to occur via the attenuation of cardiomyocyte apoptosis by the inhibition of Fas receptor-dependent apoptotic pathways [76]. It has also been indicated that the probiotic *B. animalis* subsp. *lactis* LKM512 may decrease TMA levels in healthy volunteers [77], and that probiotics may play an important role in atherosclerosis by modulating TMAO concentration [78]. 

A recent study by Montrucchio et al. [79] found serum TMAO concentrations in people living with HIV (PLWH) to be correlated with higher cardiovascular risk and atherosclerosis, and to not be affected by high-dose probiotic supplementation—no change in TMAO concentration was observed after six months of probiotic supplementation. The study included 175 participants, with TMAO concentrations of about 165 ng/mL (Table 1). More information about the reduction of TMAO level by probiotics has been described in a review by Din et al. [58].

## 6. The Effects of Probiotics, Prebiotics and Synbiotics on Uric Acid Levels

Valdivielso et al. reported the existence of a correlation between the occurrence of chronic kidney disease (CKD) and CVDs, including atherosclerosis; they also noted that the progression of atherosclerosis appeared to be related to CKD progression, as well as to a higher uric acid level [80]. An increased level of uric acid in serum has been associated with the presence of CKD and CVDs due to various molecular mechanisms, such as oxidative stress and inflammation. Conversely, certain antioxidants, such as phenolic compounds, may decrease uric acid level [81]. It has been suggested that probiotics, prebiotic and synbiotics, i.e., a combination of probiotics and prebiotics, may also change the level of uric acid. For example, Rossi et al. [82] reported that a synbiotic intervention based on the administration of 15 g of prebiotics (including a combination of three different types of fibers), and 90 billion cfu from nine different strains of *Lactobacillus, Streptococcus* and *Bifidobacteria*, daily for three weeks, may influence the outcome of CVDs and CKDs. The authors also suggested that this mixture of pre- and probiotics may reduce the levels of various toxins, including indoxyl sulphate (IS) and p-cresyl sulphate (PCS) [80]. Like TMAO, IS and PCS have been identified as proatherogenic risk factors [83].

Al-Okbi et al. [84] also noted that a combination of encapsulated probiotic (*B. bifidum*, *L. delbrueckii*, and *S. thremophilus*) and green tea alcohol extract reduces the level of uric acid in rats with hepatorenal syndrome. Wang et al. [85] reported that supplementation with *Lactobacillus* DM9218 decreases serum uric acid in fructose-fed mice; the authors suggested that it may protect against high-fructose-induced liver damage and retinol uric acid accumulation by degrading inosine. The uric acid level was also significantly reduced in non-dialysis patients with CKD who received probiotics [86]. Moreover, the eight-week consumption of *L. gasseri* PA-3 yoghurt was found to decrease the level of serum uric acid in 25 patients [87].

However, Haghighat et al. reported that daily synbiotic supplementation for 12 weeks with sachets containing 15 g of prebiotics and 5 g of probiotic (2.7 × 10^7^ cfu) did not appear to influence uric acid level in a group of 75 hemodialysis patients [58]. Garcia-Arrayo et al. [88] found that five-week probiotic supplementation with *L. acidophilus* KB27 (5 billion cfu/daily) and *L. rhamnosus* KB79 (5 billion cfu/daily) elevates uric acid extraction in rats; it was also found to increase oxonic acid- induced intrarenal accumulation, which is known to stimulate hyperuricemia and renal damage. Firouzi and Haghighatdoost [89] reported elevated uric acid levels in healthy people (*n* = 437) who consumed probiotics, prebiotics and synbiotics, compared with a placebo group.

## 7. Conclusions

Although it has been proposed that probiotic consumption may bestow anti-hypertensive properties and be associated with improved serum lipid profiles [57,58,90,91], these postulated benefits were based on an in vitro model. More recent in vivo trials indicate that the consumption of probiotics, prebiotics or a combination of the two, i.e., synbiotics, may assist in the prevention and treatment of cardiovascular diseases; however, few studies have examined the role of probiotics in this regard, and their precise medicinal and prophylactic doses for humans remain unknown (Table 1). It is also very difficult to demonstrate the specific correlation, relationship and synergic relationships between probiotics, prebiotics and synbiotics in the prevention and treatment of CVD. However, Liong et al. [57,59] and Haghighat et al. [59] indicated that probiotics, prebiotics and synbiotics may offer promise in the prevention and treatment of CVD; for example, a synbiotic containing *L. acidophilus* ATCC 4962 and three commercially available prebiotics (mannitol, fructooligosacharides and inulin) has been found to reduce total cholesterol, triacylglycerol and LDL-cholesterol levels (Table 1). Moreover, other authors [21] describe the role of probiotics in patients with stable coronary artery disease. On the other hand, their role in patients with other CVDs, including stroke, cardiomyopathy, and carditis, is not known.

In addition, the mechanisms of action of probiotics, prebiotics and synbiotics remain poorly defined. Probiotics are believed to modulate various factors which may play an important role in CVDs (Figure 3). Similarly to prebiotics and synbiotics, they may lower cholesterol levels by increasing bile salt synthesis and bile acid deconjugation, and may offer protection from cardiovascular diseases. Moreover, probiotics have been found to have anti-oxidative, anti-platelet and anti-inflammatory properties (Figure 3). However, further experimental research, particularly well-designed clinical studies, are needed before these substances can effectively be used in the prevention and treatment of CVD. Finally, no studies have yet evaluated the effect of their use on CVDs in combination with classical therapy, for example aspirin; therefore, although no side-effects have been reported in clinical studies to date, it cannot be unequivocally stated whether probiotic, prebiotic and synbiotic use represents an effective strategy for the prevention and treatment of CVDs. 

## Figures and Tables

**Figure 1 ijms-21-09737-f001:**
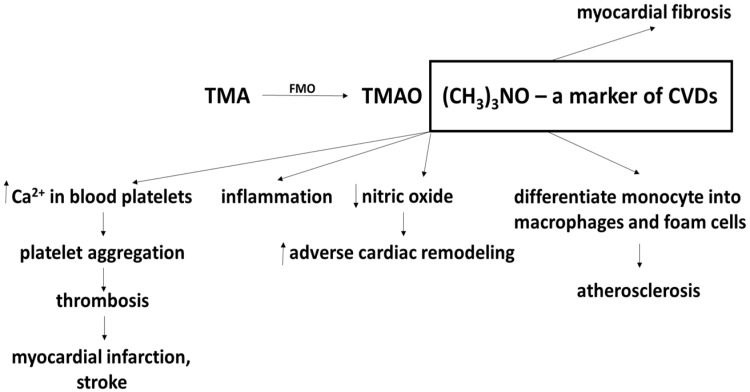
Effect of trimethylamine-N-oxide (TMAO) on cardiovascular diseases (CVDs) (modified) [70] More details in text.

**Figure 2 ijms-21-09737-f002:**
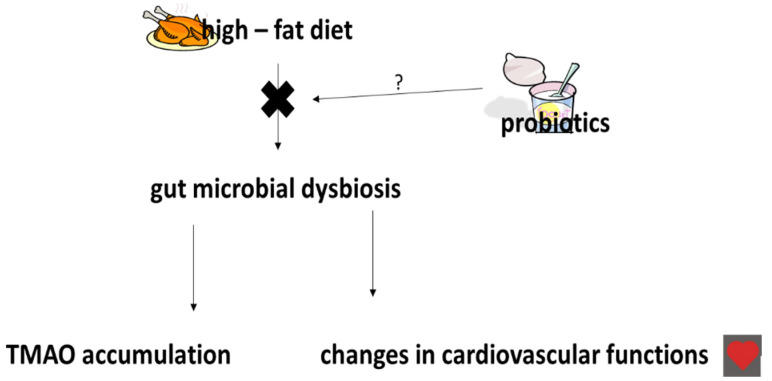
Effect of probiotics on trimethylamine-N-oxide (TMAO) accumulation and changes in cardiovascular functions. More details in text (modified) [69].

**Figure 3 ijms-21-09737-f003:**
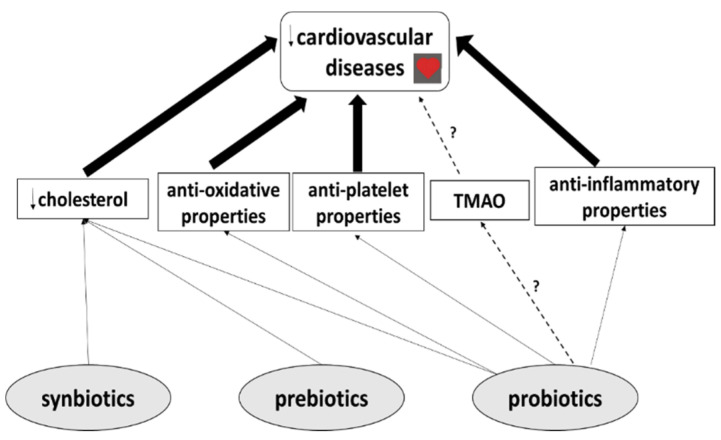
The effect of probiotics, prebiotics and synbiotics on cardiovascular disease. Probiotics lower the level of cholesterol by increasing bile salt synthesis and bile acid deconjugation. Probiotics also have anti-oxidative, anti-platelet and anti-inflammatory properties, and hence may offer protection from cardiovascular diseases. One proposed model of preventing the CVD-induced accumulation of TMAO may be probiotic supplementation. However, prebiotics and synbiotics may also decrease the level of total cholesterol and offer similar protection from cardiovascular disease. More details in text.

**Table 1 ijms-21-09737-t001:** The effect of probiotics, prebiotics, and synbiotics on various risk factors of cardiovascular diseases (*in vivo* model).

Probiotic/Prebiotic/Synbiotic Consumption	Dose (Per Day)	Days/Weeks	Subjects	Risk Factors of Cardiovascular Diseases	Ref.
Probiotic					
*L. plantarum* 299v	20 billion cfu	6 weeks	Patients with stable coronary artery disease (people)	Reducing inflammatory biomarkers. No effect on trimethylamine-N-oxide (TMAO) level	[21]
*L. plantarum* PH04	10^7^ cfu	14 days	Male mice fed a high-cholesterol diet containing 10% *w*/*v* skim milk and 10% *w*/*v* cream	Lowering total cholesterol and triglycerides	[23]
*L. reuteri* V3401	5 × 10^9^ cfu	12 weeks	Obese adults (people) with metabolic syndrome	Reducing inflammation	[30]
Multispecies probiotic Ecologic^®^ Barrier	2.5 × 10^9^ cfu; 1 × 10^10^ cfu	12 weeks	Obese postmenopausal Caucasian women	Decreasing systolic blood pressure and inflammation	[31]
Multispecies probiotic Ecologic^®^ Barrier	2.5 × 10^9^ cfu	12 weeks	Obese women	Decreasing homocysteine	[43]
*L. acidophilus*	10^9^ cfu/mL	4 weeks	Hypercholesterolemic rats	Lowering total cholesterol and low-density lipoprotein (LDL)	[24]
*L. fermentum* MTCC:5898-fermented milk	2 × 10^9^ cfu	90 days		Reducing dyslipidemia, oxidative stress and inflammation	[29]
*L. rhamnosus* LRH11 and *L. plantarum* SGL07	3 × 10^8^ cfu	16 weeks	People with CVDs risks	Reducing plasma trimethylamine-N-oxide (TMAO) level	[75]
*B. animalis* subsp. *lactis* LKM512	No described	12 weeks	Healthy people	Reducing plasma trimethylamine-N-oxide (TMAO) level	[77]
*L. casei* Shirota	6.5 × 10^9^ cfu	12 weeks	Subjects with metabolic syndrome	No effect on trimethylamine-N-oxide (TMAO) level	[72]
*S. thermophilis, L. acidophilus* LA-5 and *B. bifidum* BG-12	No described (3 mL/day)	4 weeks	Healthy animals	Lowering cholesterol and triglyceride	[20]
Prebiotic					
Prebiotic fiber	10 or 20%	10 weeks	Obese hyperlipidemic rats	Lowering total serum cholesterol and triacylglycerol	[50]
Oligofructose	8 g/day	16 weeks	Overweight and obese children aged 7–12 years	Decreasing body weight	[53]
Oligofructose	21 g/day	12 weeks	Overweight and obese adults	Decreasing body weight	[51]
Synbiotic					
Prebiotic (three different fiber types: 5 g fructooligosaccharides, 5 g galactooligosaccharides and 5 g inulin) and probiotic powder (Bioflora^®^) containing *L. acidophilus* strain T16, *B. bifidum* strain BIA-6, *B. lactis* strain BIA-6, and *B. longum* strain LAF-5	15 g of prebiotic and 5 g probiotic in sachet	12 weeks	Hemodialysis patients	Reducing the concentration of intracellular adhesion molecule type 1	[59]
Synbiotic containing *L. acidophilus* ATCC 4962, and three commercially available prebiotics (mannitol, fructooligosacharides and inulin)	*L. acidophilus* ATCC 4962 (1 g/pig per day), mannitol (1.56 g/pig per day), fructooligosacharides (1.25 g/pig per day) and inulin (2.2 g/pig per day)	8 weeks	Hypercholesterolemic pigs	Reducing total cholesterol, triacylglycerol, low-density lipoprotein (LDL)-cholesterol	[57,58]

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
