# Peer review of "Probiotics, Prebiotics and Synbiotics—A Promising Strategy in Prevention and Treatment of Cardiovascular Diseases?"

_ijms, 2020, doi:10.3390/ijms21249737_

Round 1
Reviewer 1 Report
The review ijms-1014605 report studies regarding the use of probiotics, prebiotics and synbiotics as dietary components in the prevention and treatment of cardiovascular diseases (CVD). In its overall the review could be interesting but some of the references are overdate. I suggest to consider only recent studies (2015-2020).
The author mention in the review several times the limited number of studies, especially in vivo, related to the topic. Could you give an explanation? There are plenty of studies related to effect of probiotics, prebiotics and synbiotics on the human health, but less interest on the CVD prevention and therapy, why?
I suggest the review for publication after major changes;
1) revision of the literature, add more recent studies;
2) the review does not address a particular message. Even if, according to the author, there are not so many studies, the meaning of the review should be labelled
2) when in vivo studies are mentioned, the population in which have been carried out should be mentioned
3) The quality of the Figures has to be improved. Figure 2 can be removed
4) Make a table in a better format. A 4 pages table is not attractive
5) funding and acknowledgements are not reported. Does the author have?
Author Response
The review ijms-1014605 report studies regarding the use of probiotics, prebiotics and synbiotics as dietary components in the prevention and treatment of cardiovascular diseases (CVD). In its overall the review could be interesting but some of the references are overdate. I suggest to consider only recent studies (2015-2020).
I would like to thank the Reviewer for providing helpful comments. The manuscript has been revised accordingly with the adjusted text being given in red.
Response: I have not corrected the references, because papers (<2015) are also very important, for example:
Pandey KR, Naik SR, Vakil BV. Probiotics, prebiotics and synbiotics – a review. J. Food Sci Technol 2010;52:7577-87.
Sudha MR, Chauhan P, Dixit K, Babu S, Jamil K. Probiotics as complementary therapy for hypercholesterolemia. Biol Med 2009;1:1-13.
Moreover, I have added new papers (2020). For example:
Chi C, Li C, Wu D, Buys N, Wang W, Fan H, Sun J. Effects of probiotics on patients with hypertension: a systematic review and meta-analysis. Curr Hypertens Rep 2020;21:1-12.
Shah BR, Li B, Al Sabbah H, Xu W, Mraz J. Effects of prebiotic dietary fibers and probiotics on human health: with special focus on recent advancement in their encapsulated formulations. Trends Food Sci Technol 2020;102:178-92.
The author mention in the review several times the limited number of studies, especially in vivo, related to the topic. Could you give an explanation? There are plenty of studies related to effect of probiotics, prebiotics and synbiotics on the human health, but less interest on the CVD prevention and therapy, why?
Response: Recent evidence suggests that probiotics, prebiotics and synbiotics may serve as important dietary components in the prevention (especially) and treatment of cardiovascular diseases (CVD), but the recommendations for their use are often based on brief reports and small clinical studies. This review evaluates the current literature on the correlation between CVD and probiotics, prebiotics and synbiotics. Although research on probiotics, prebiotics and synbiotics has grown exponentially in recent years, particularly regarding the effect of probiotics on CVD, their mechanisms have not been clearly defined. has recently grown exponentially. For example, 201 articles were published during 2014-2020, and 63 were published on “probiotics and cardiovascular diseases” during 2009-2013 (PubMed June, 2020). Therefore, the aim of this review is to discuss the roles of probiotics, prebiotics and synbiotics in the prevention and treatment of cardiovascular diseases based on data from journals recorded in international databases, including inter alia PubMed and Scopus.
I suggest the review for publication after major changes;
- revision of the literature, add more recent studies;
Response: I have not corrected the references, because papers (<2015) are also very important, for example:
Pandey KR, Naik SR, Vakil BV. Probiotics, prebiotics and synbiotics – a review. J. Food Sci Technol 2010;52:7577-87.
Sudha MR, Chauhan P, Dixit K, Babu S, Jamil K. Probiotics as complementary therapy for hypercholesterolemia. Biol Med 2009;1:1-13.
Moreover, I have added new papers (2020). For example:
Chi C, Li C, Wu D, Buys N, Wang W, Fan H, Sun J. Effects of probiotics on patients with hypertension: a systematic review and meta-analysis. Curr Hypertens Rep 2020;21:1-12.
Shah BR, Li B, Al Sabbah H, Xu W, Mraz J. Effects of prebiotic dietary fibers and probiotics on human health: with special focus on recent advancement in their encapsulated formulations. Trends Food Sci Technol 2020;102:178-92.
- the review does not address a particular message. Even if, according to the author, there are not so many studies, the meaning of the review should be labelled
Response: I have modified the chapter of Introduction and Conclusion. “Dietary supplements, often termed functional foods, can bestow beneficial effects against various risk factors associated with cardiovascular diseases. However, little, if anything, is known about the role of probiotic, prebiotic and symbiotic supplements as important dietary components in the prevention and treatment of CVD. For example, in review paper, Vascquez et al. (10) indicate that probiotics may decrease the production of reactive oxygen species, and reduce oxidative stress. Nevertheless, some brief reports and small clinical studies have been conducted, and these may nevertheless offer new directions in treatment; in addition, the body of research regarding these supplements, particularly concerning the effect of probiotics on cardiovascular diseases, has recently grown exponentially. For example, 201 articles were published during 2014-2020, and 63 were published on “probiotics and cardiovascular diseases” during 2009-2013 (PubMed June, 2020). Therefore, the aim of this review is to discuss the roles of probiotics, prebiotics and synbiotics in the prevention and treatment of cardiovascular diseases based on data from journals recorded in international databases, including inter alia PubMed and Scopus.”; “In addition, the mechanisms of action of probiotics, prebiotics and synbiotics remain poorly defined. Probiotics are believed to modulate various factors which may play an important role in CVDs (Figure 3). Similarly to prebiotics and synbiotics, they may lower cholesterol levels by increasing bile salt synthesis and bile acid deconjugation, and may offer protection from cardiovascular diseases. Moreover, probiotics have been found to have anti-oxidative, anti-platelet and anti-inflammatory properties (Fig. 3). However, further experimental research, particularly well-designed clinical studies, is needed before these substances can effectively be used in the prevention and treatment of CVD. Finally, no studies have yet evaluated the effect of their use on CVDs in combination with classical therapy, for example aspirin; therefore, although no side-effects have been reported in clinical studies to date, it can not be unequivocally stated whether probiotic, prebiotic and synbiotic use represents an effective strategy for the prevention and treatment of CVDs.”
3) The quality of the Figures has to be improved. Figure 2 can be removed
Response: I have corrected figures and their legends. However, I have not deleted Figure 2, because preliminary results suggest that probiotic treatment may prevent the development of CVDs following TMAO accumulation (Fig. 2).
- Make a table in a better format. A 4 pages table is not attractive
Response: I have corrected Table.
5) funding and acknowledgements are not reported. Does the author have?
Response: I have not funding and acknowledgements
Reviewer 2 Report
I would reccomend to check and modify the figures and the table, words are to big and somefigures have words over the draws that make it difficult to understand
Regarding the conclusion I would make it shorter
Please check the references some italics should be changed
Author Response
I would reccomend to check and modify the figures and the table, words are to big and somefigures have words over the draws that make it difficult to understand
I would like to thank the Reviewer for providing helpful comments. The manuscript has been revised accordingly with the adjusted text being given in red.
Response: I have corrected figures and their legends and the table.
Regarding the conclusion I would make it shorter
Response: I have shorted the conclusion. Now, it is: “In addition, the mechanisms of action of probiotics, prebiotics and synbiotics remain poorly defined. Probiotics are believed to modulate various factors which may play an important role in CVDs (Figure 3). Similarly to prebiotics and synbiotics, they may lower cholesterol levels by increasing bile salt synthesis and bile acid deconjugation, and may offer protection from cardiovascular diseases. Moreover, probiotics have been found to have anti-oxidative, anti-platelet and anti-inflammatory properties (Fig. 3). However, further experimental research, particularly well-designed clinical studies, is needed before these substances can effectively be used in the prevention and treatment of CVD. Finally, no studies have yet evaluated the effect of their use on CVDs in combination with classical therapy, for example aspirin; therefore, although no side-effects have been reported in clinical studies to date, it can not be unequivocally stated whether probiotic, prebiotic and synbiotic use represents an effective strategy for the prevention and treatment of CVDs.”
Please check the references some italics should be changed
Response: I have corrected the references.
Round 2
Reviewer 1 Report
accept for publication